# LARGE-SCALE PRETRAINING FOR NEURAL MACHINE TRANSLATION WITH TENS OF BILLIONS OF SENTENCE PAIRS

## ABSTRACT

In this paper, we investigate the problem of training neural machine translation (NMT) systems with more than 40 billion bilingual sentence pairs, which is orders of magnitude larger than the largest dataset to date. Several challenges distinguish this problem from previous NMT works such as the extreme noisiness of the data and prohibitively long training time. We propose practical solutions to handle these issues and demonstrate that large-scale pretraining will significantly improve NMT performances. By merely using vanilla transformer structures, we are able to push the BLEU score of WMT17 Chinese-English dataset to 32.3, with a significant performance boost of +3.2 over existing SOTA results, along with achieving SOTA results on WMT18 (+2.6) and WMT19 (+2.9).

## 1 INTRODUCTION

End-to-end neural machine translation (NMT) (Bahdanau et al., 2014; Sutskever et al., 2014; Luong et al., 2015; Sennrich et al., 2015b; Vaswani et al., 2017; Britz et al., 2017; Gehring et al., 2017; Klein et al., 2017; Johnson et al., 2017; Wang et al., 2017; Hassan et al., 2018; Aharoni et al., 2019; Ng et al., 2019) has been widely adopted as the state-of-the-art approach for MT. For bilingual MT, sequence-to-sequence models (SEQ2SEQ for short) are trained and tested on publicly available benchmarks, the size of which ranges from tens of thousand for low-resource languages to tens or hundreds of million for widely used languages.

Recent progress in natural language understanding (NLU) has proved that large-scale pretraining such as BERT (Devlin et al., 2018) or Elmo (Peters et al., 2018) will lead to a significant leap-forward in SOTA results. Mostly due to the lack of bilingual training data (Song et al., 2019) no comparable success has been made in the field of MT, and two important questions have mostly remained unanswered: WHETHER and HOW we can push the performance of existing neural models if we increase the amount of bilingual data by orders of magnitude, e.g., if we are presented with ten of billions of bilingual training sentence pairs. The answers to these questions are not immediately clear and several aspects distinguish this problem from all previous NMT works:

- Scale: Firstly, an NMT model's expressivity is limited by infrastructures such as GPU memory, and indefinitely increasing the size of the training data might not improve the performance. Secondly, training on a massively large dataset with tens of billion sentence pairs can be prohibitively slow.
- Noise and Out-of-domain data: A dataset with tens of billions of bilingual sentences pairs must span a wide range of domains and comes from extremely noisy resources. It is widely accepted that translation quality is very vulnerable to out-of-domain data and noisy data (Chen et al., 2016; Niehues & Waibel, 2010; Koehn & Schroeder, 2007; Eidelman et al., 2012). and small amounts of noisy training instances can have negative effects on translation quality (Belinkov & Bisk, 2017). This means blindly increasing the size of training data or adding noisy training data may not necessarily lead to better performances, but may lead to worse performances.

In this paper, we investigate the problem of training neural machine translation systems on a dataset with more than 40 billion bilingual sentence pairs, which is orders of magnitude larger than the

largest dataset to date. To tailor existing WMT systems to the massively large training dataset, we propose a pipelined strategy which involves large-scale pretraining and domain-specific fine-tuning. We handle the trade-off between full-dataset optimality and convergence speed for pretraining and demonstrate that the combination of large-scale pretraining will significantly improve NMT performances. Combing other existing NMT techniques (Hassan et al., 2018; Ranzato et al., 2015; Li et al., 2016), we are able to push to the BLEU score of WMT 2017 Chinese-English translation to 32.2, with a significant performance boost of +3.1 over existing SOTA results.

## 2 RELATED WORK

Large scale pretraining has proved to be of significant importance in NLP, from the early time of word vectors such as word2vec or Glove (Pennington et al., 2014; Mikolov et al., 2013) to recent language model pretraining such as BERT (Devlin et al., 2018) or Elmo (Peters et al., 2018). Pretraining has not achieved comparable success in MT. This is mostly because of the difficulty to get a large parallel corpus. Most NMT pretraining work focuses on unsupervised NMT (Lample & Conneau, 2019) or pretraining using monolingual data (Song et al., 2019). Other relevant works for SEQ2SEQ pretraining include using an auto-encoder to pretraining the encoder-decoder network (Dai & Le, 2015; Ramachandran et al., 2016) and transfer learning from rich-source languages to low-source languages (Zoph et al., 2016; Firat et al., 2016).

Our work is related to a group of previous work in domain adaptation for MT. In the context of phrase-based MT, Hildebrand et al. (2005) selects sentences similar to the topic of the test set to construct a new training corpus to avoid topic discrepancies between the training and test datasets. Xiao et al. (2012) estimate word translation probabilities conditioned on topics, and then adapt lexical weights of phrases by these topic-specific probabilities. In the context of NMT, Chen et al. (2016) feeds neural networks with topic information (which on human-labeled product categories) on the decoder side; Zhang et al. (2016b) first runs topic models (Blei et al., 2003) on the training data (both sources and targets), and incorporate topic representations into the encoder-decoder model.

Mixture models have been widely studied used in MT. In the context phrase-based MT. Foster & Kuhn (2007) propose a three-step pipelined strategy which first splits, the training corpus into different sub-corpora according to some predefined criterion, then trains different MT models on different sub-corpora, and in the end combine different models for translation. Foster & Kuhn (2007)'s work lated was extended in subsequent work (Niehues & Waibel, 2010; Koehn & Schroeder, 2007; Eidelman et al., 2012), regarding different constituent aspects such as how to split the training corpus (Axelrod et al., 2011) and how to combine different models (Civera & Juan, 2007; Sennrich, 2012; Foster et al., 2010). In NMT, mixture models (Shen et al., 2019; He et al., 2018) are inspired by deep latent variable generation models (Kingma & Welling, 2013; Kim et al., 2018; Bowman et al., 2015). Zhang et al. (2016a) augment NMT systems with a single Gaussian latent variable, and this work was further extended by Schulz et al. (2018) which associated each target word with a latent Gaussian variable. He et al. (2018) propose to use a soft mixture model to generate diverse translations. Further, Shen et al. (2019) comprehensively evaluate different design choices of mixture models such as parameterization and prior distribution.

Due to the massively large dataset, we have to split it for training. This makes this paper relevant to a wide range of previous work on distributed training for deep networks (Dean et al., 2012; Yadan et al., 2013; Li et al., 2014; Krizhevsky, 2014; Das et al., 2016; Smith et al., 2017).

## 3 DATA SETUP

The most commonly used Chinese-to-English (Zh-En) translation dataset is the WMT'17. The dataset consists of 332K sentence pairs from the News Commentary corpus, 15.8M sentence pairs from the UN Parallel Corpus, and 9M sentence pairs from the CWMT Corpus. We followed the pre-processing criterion in (Hassan et al., 2018), resulting in about 20M training pairs. Newsdev2017 is used as the development set and Newstest2017 as the test set.

In addition to WMT2017, we collected a Chinese-English parallel dataset that significantly extends those studied by previous works. The original dataset consists of roughly 50 billion sentence pairs in total. The data comes from from diverse sources such as web pages ($\sim$2 billion), digitized books ($\sim$1 billion) and private purchase from translation agencies ($\sim$46 billion).

The dataset is extremely noisy in two ways: (1) significant proportion of files are of the PDF format. (2) texts are aligned at the document level rather than sentence level. The size of large documents can be up to thousands of pages, with figures, tables and charts annoyingly inserted. For the first part, we developed a PDF document parsing system to decode PDF files. Since this part are out of the scope of this paper, we omit the details. Succinctly, a lexical analyzer is first used to decompose PDF files into the basic lexical tokens according to PostScript syntax. Next, we build a parser to analyze the abstract syntax tree (AST), and then decode the data into figures, tables, and texts. For the second part, our goal is to transform doc-level alignments to sentence-level alignments. We use a hierarchical pipeline which consists of two stages: (1) aligning paragraphs and (2) aligning sentences within each aligned paragraph. We adopted many of the techniques in Uszkoreit et al. (2010): Paragraphs/sentences are discarded if both sides are identical or a language detector declares them to be in the wrong language. We used a standard dynamic programming sentence/paragraph alignment algorithm using sentence length and translation probability as features. Pairs are discarded if pairing scores are less than a certain threshold. We encourage the readers to refer to (Uszkoreit et al., 2010) for details. After post-processing, we are left with 41 billion sentence pairs. We randomly select 1M instances as the development set.

## 4 MODELS AND ARCHITECTURES

### 4.1 PIPELINES: PRETRAINING AND FINE-TUNING

Translation quality is very vulnerable to out-of-domain and noisy data (Chen et al., 2016; Niehues & Waibel, 2010; Koehn & Schroeder, 2007; Eidelman et al., 2012). Since the dataset we use comes from very noisy resources and covers a wide range of domains, it would be less favorable if we directly apply a trained model on this large but noisy dataset to the test set. One option to handle this issue is do data selection and filtering before training (Belinkov & Bisk, 2017; Hassan et al., 2018). Hassan et al. (2018) proposes to first learn sentence representations from the provided training data in WMT2017 (target domain), and then reject training instances if their similarity with below a specified threshold. We did not choose this choice for two reasons: (1) Hassan et al. (2018) selects different training instances for different target domains. This means every time we encounter a new domain, we have to retrain the model; (2) the value of data filtering threshold is crucial but hard to decide: it is unrealistic to tune its value since each threshold value corresponds to a different filtered training set, on which a brand new model has to be trained.

Inspired by large-scale pretraining strategies such as BERT (Devlin et al., 2018) or Elmo (Peters et al., 2018), we used a pipelined approach: we first pretrain an NMT model on the massively large dataset, and then fine-tune the model on the training set of the target domain. This strategy naturally addresses the aforementioned two caveats of data pre-filtering approach: the pretrained model can be easily adapted to a set of training data of an arbitrary domain, and there is no issue of data selection threshold. Moreover, since the model will be fine-tuned in the latter stage, it is more immune to the data noisiness in the first stage. We used the combination of WMT 20M data and 40B data for pretraining, and then fine-tune the model on the 20M data.

### 4.2 TRADEOFF BETWEEN FULL-DATASET OPTIMALITY AND CONVERGENCE SPEED

Optimality on the full dataset trades convergence speed: one end of the spectrum is to run a single model on the full dataset to find full-dataset optimality. This strategy suffers prohibitively long training time.[1] The other end of the spectrum is to split the full dataset into smaller subsets and run independent models on different datasets until convergence, as in (Foster & Kuhn, 2007). In this case, each individual model finds the optimal for the subset, but their combination is not guaranteed to be the optimal for the full set. The middle point of the spectrum involves model communication or data communication.

---

[1] With a single V100 GPU, a single update on a single batch with about 2-5K tokens takes about 2-6 seconds. Even with 512 parallel GPUs, it takes months for an epoch to finish.

### 4.3 Model Details

We use the Transformer architecture (Vaswani et al., 2017) as a backbone. The encoder and decoder have 6 blocks. The number of attention heads, embedding dimension and inner-layer dimension are set to 16, 1,024 and 2,048. We use the same transformer structure for all experiments. All experiments are run using 512 Nvidia V100 GPUs with mini-batches of approximately 1M tokens. Models are optimized with Adam (Kingma & Ba, 2014) with $\beta_1$ set to 0.9, $\beta_2$ set to 0.98 and $\epsilon$ set to $10^{-8}$. On the Chinese side, instead of using segmented words or byte-pair encoding (BPE) (Sennrich et al., 2015b), we use characters as the basic units and maintain a character vocab size of 10,000. We will get back to this in the ablation study section. On the English side, we segmented texts into subword symbols using BPE (Sennrich et al., 2015b) and maintain a vocab size of 40K.

#### 4.3.1 Pretraining

We explore the following different strategies for model pretraining.

**single-model**  We use a single transformer model to fit all the training data. We use 512 Nvidia V100 GPUs with mini-batches of approximately 1M tokens. Models are saved every 0.1 epoch. Upon the publication of this paper, training has lasted for three months, 2 epochs in total, and perplexity on the dev set is still dropping.

**uniform-data-split**  the disadvantages of *single-model* are obvious: (1) It is prohibitively slow and (2) it is unclear whether a single-model is powerful enough to express the full training dataset. We thus followed Foster & Kuhn (2007) to build mixture models, in which we first split the full dataset into a few subsets, and then training independent model components[2] on different subsets. Using multiple components naturally increase the model's capacity and expressivity.

We randomly split the 40B training set into $K = 10$ subsets, denoted by $D_1, D_2, ..., D_K$. At training, different transformers are independently trained on different subsets using parallel GPUs until convergence. At test time, the probability of $p(y|x)$ can be written as follows:

$$p(y|x) = \sum_z p(y|z, x)p(z|x) \tag{1}$$

where $z$ can be simply thought as the index of different subsets. $p(y|z, x)$ is characterized by the SEQ2SEQ component trained on $D_z, z \in [1, K]$. We assume that $p(z|x) = 1/K$ is uniform. The generation of target $y$ is thus the ensemble of the $K$ models.

**topic-data-split**  The issue with *uniform-data-split* is that the subsets are randomly generated. We wish that each $D_z$ represents a specific domain and that components trained on different $D_z$ are separate apart and have their own specializations or domain of expertise. Domain-specific $D_z$ come with the advantages of fewer vocabularies, fewer low-frequency words and sentences of more similar topics and language expression patterns. We thus propose to split the full dataset in a more elegant way using topic models (Blei et al., 2003). Topic models were widely used for data split and selection at the time of phrase-based MT (Hildebrand et al., 2005; Zhao & Xing, 2006; 2008). One tricky issue here is that each sentence pair consists of two different languages and we need to extract bilingual topics. To handle this issue, Zhao & Xing (2008) proposed a generative model, in which each source sentence is first sampled based on its topic, as in standard LDA. Then for each position of the source sentence, a target word is sampled based on a topic specific translation lexicon. Variational EM is used for inference. We refer the readers to Zhao & Xing (2008) for details. We followed Zhao & Xing (2008) and mined topics distribution from the bilingual corpus. Each sentence pair is assigned to the subset to which it belongs with the largest probability.

When data split is done, different SEQ2SEQ components are independently trained on different $D_z$. At test time, we use $p(y|x) = \sum_z p(y|z, x)p(z|x)$ for inference. $p(y|z, x)$ is characterized by the SEQ2SEQ component trained on $D_z$. Unlike *uniform-data-split*, $p(z|x)$ is not uniform, but is predicted by a $K$-class classification model using BiLSTMs. The classification model first maps $x$ to a vector representation and outputs the representation to a $K$-class softmax function.

---

[2]We use the term "component" to denote individual transformers in the mixture model setup.

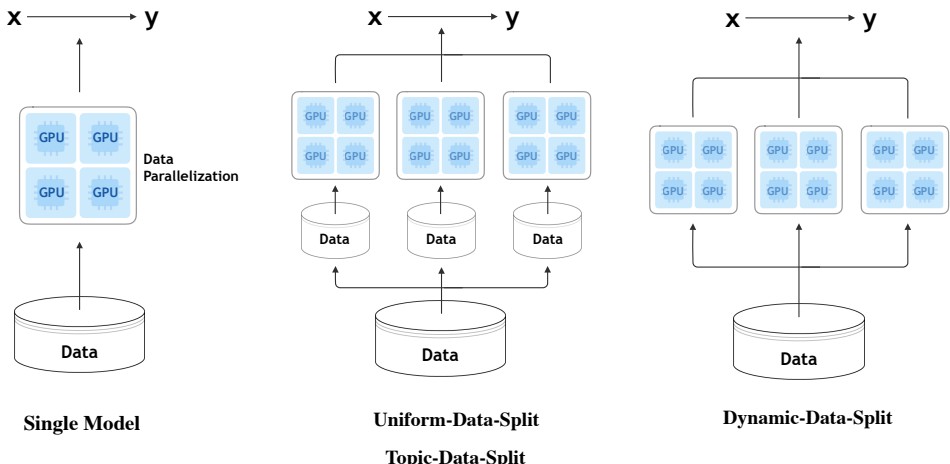

Figure 1: Overview of different strategies for training.

**dynamic-data-split**    For *topic-data-split* and *uniform-data-split* strategies, subsets are generated beforehand and then fixed during training. We are thus not able to dynamically adjust data during training. This means that if a training pair is erroneously clustered into a subset that it does not belong to, it will negatively affect the training process permanently. Inspired by mixture models for NMT (Shen et al., 2019; He et al., 2018), we propose to dynamically assign training instances to different components, and update different components according to the examples assigned.

$$\nabla_\theta \log \sum_z p(y|z,x)p(z|x) = \mathbb{E}_{p(z|x,y)}\nabla_\theta \log p(y,z|x) \qquad (2)$$

We choose to use hard-EM instead of vanilla-EM because of the concern about training speed: for vanilla-EM, all components need to run a forward-backward step on each training instance.[3] This is computationally intensive and we will again be faced with slow convergence issue. Using hard-EM, models are trained by iteratively applying the following two steps:

**E-step**: For a given sentence pair $(x_i, y_i)$, estimate $z_i$ by: $z_i = \text{argmax}_z \log p(y_i|z_i, x_i)$

**M-step** update parameters with gradients $\nabla_\theta \log p(y_i, z_i|x_i)$

We need to extend the single-instance EM step above to batched computation. There are two issues that we need to specially care about: (1) the nortorious "richer gets richer" issue (Eigen et al., 2013; Shazeer et al., 2017; Shen et al., 2019): once one component is more adequately trained than others, it is always picked and other components will never get trained. Then the latent variable $z$ becomes useless and the entire system degenerates to the vanilla SEQ2SEQ model modeling $p(y|x)$; (2) to accelerate training, we wish all components keep getting updated and trained. Recall that each component actually consists of hundreds of parallel GPUs. Components not getting enough data would be extremely resource-wise inefficient.

We propose the **batched-E-step** strategy to deal with these two issues: suppose that the batch size for mixture model component is $B$ (the value of which is approximately 1M). Since we have $K$ components, we feed $K \times B$ training instances to the model for each E-step. We propose that each component is **guaranteed** to be assigned $B$ instances in the E-step in oder to get sufficiently updated

---

[3]For different values of $z_i$, each $p(y_i|z_i, x_i)$ will be updated by gradients $\nabla_\theta \log p(y_i, z_i|x_i)$ weighted by corresponding responsibility $p(z_i|x_i)$.

| Training Data | Setting | Performance |
|---|---|---|
| | WMT2017 | |
| 20M | Transformer (Hassan et al., 2018) | 24.4 |
| 20M (ensemble) | Transformer | 26.7 |
| 20M | Sogou (Wang et al., 2017) | 26.4 |
| 20M | Microsoft (Hassan et al., 2018) | 27.4 |
| 20M (ensemble) | Teacher Forcing (He et al., 2019) | 29.1 |
| 20M+100M (ensemble) | Microsoft (Hassan et al., 2018) | 28.4 |
| 20M+40B (pretrain only) | single-model (1 epoch) | 22.1 |
| 20M+40B (pretrain only) | single-model (2 epoch) | 24.7 |
| 20M+40B (pretrain only) (ensemble) | Uniform Data Split | 27.2 |
| 20M+40B (pretrain only) (ensemble) | Topic Data Split | 27.7 |
| 20M+40B (pretrain only) (ensemble) | Dynamic Data Split | 28.4 |
| 20M+40B (pretrain+finetune) | single-model (1 epoch) | 27.4 |
| 20M+40B (pretrain+finetune) | single-model (2 epoch) | 28.7 |
| 20M+40B (pretrain+finetune) (ensemble) | Uniform Data Split | 31.1 |
| 20M+40B (pretrain+finetune) (ensemble) | Topic Data Split | 31.5 |
| 20M+40B (pretrain+finetune) (ensemble) | Dynamic Data Split | 32.0 |

Table 1: Main results of different models and different settings on the WMT 2017 Chinese-English test set.

in the following M step. The batch-level assignment $z$ are computed as follows:

$$z_1, z_2, ..., z_{BK} = \underset{z_1, z_2, ..., z_{BK}}{\arg\max} \sum_{i=1}^{i=BK} \log p(y_i | z_i, x_i)$$

$$\text{subject to} \sum_{i=1}^{BK} \mathbb{I}(z_i = z) = B \ \text{ for } z = 1, 2, ..., K \tag{3}$$

Eq. 3 is an integer linear programming (ILP) problem. ILP is NP-hard and is solved using Hill Climbing Russell & Norvig (2016), a heuristic method to find the optimal. The proposed strategy naturally simultaneously addresses the aforementioned "richer gets richer" issue and machine-not-being-used issue. It is worth noting that the proposed **batched-E-step** is not specific to our scenario, but provides a general solution to the long-standing degeneracies of neural mixture models for text generation (Shazeer et al., 2017; Shen et al., 2019).

### 4.3.2 FINE-TUNING

For the model fine-tuning stage, we maintain the structure of the original pretrained model and fine-tune the model on the 20M WMT17 dataset. The number of iteration is treated as a hyper-parameter to tune on the dev set of WMT17.

For *single-model*, we maintain the transformer structure and run additional iterations. For *uniform-data-split* and *topic-data-split*, we fine-tune each component on the WMT17 dataset. At test time, constituent components are combined for decoding. Translations from *uniform-data-split* and *topic-data-split* are generated by the ensemble of $K$ models.

For *dynamic-data-split*, we proceed to run mixture models on the WMT17 dataset with minor adjustments: we replace the hard-EM and the batched-E-step with vanilla soft-EM, in which all components get updated with each training instance. We do this because: (1) The WMT17 dataset is significantly smaller and we don't have to worry about the computational cost; (2) since mixture models have already been sufficiently trained during pretraining, we are less concerned with the "richer gets richer" issue. Or even, we wish that some components get sufficiently fine-tuned if they are relevant to the target domain, while others remain dormant.

## 5 EXPERIMENTAL RESULTS

WMT17 results are reported in Table 1. WMT18 and 19 results are shown in Table 2.

| Setting | Performance |
|---|---|
| WMT 2018 | |
| Cambridge (Stahlberg et al., 2018) (ensemble) | 27.7 |
| Tencent (Wang et al., 2018) (ensemble) | 29.3 |
| Microsoft (Xia et al., 2019) (ensemble) | 30.9 |
| Baidu (Sun et al., 2019) (ensemble) | 31.8 |
| Uniform Data Split (ensemble) | 32.8 |
| Topic Data Split (ensemble) | 33.5 |
| Dynamic Data Split (ensemble) | 34.4 |
| WMT 2019 | |
| NiuTrans (Li et al., 2019) (ensemble) | 34.2 |
| Baidu (Sun et al., 2019) (ensemble) | 38.0 |
| Microsoft (Xia et al., 2019) (ensemble) | 39.3 |
| Uniform Data Split (ensemble) | 41.0 |
| Topic Data Split (ensemble) | 41.6 |
| Dynamic Data Split (ensemble) | 42.2 |

Table 2: Pretrain+Finetuning results on WMT 2018 and WMT2019 Chinese-English test set.

## 5.1 Results

**Baselines** We copied baseline results from the Sogou best WMT17 system (Wang et al., 2017), the microsoft system (Hassan et al., 2018), and the current SOTA result using teaching forcing (He et al., 2019).

**Pretrain Only** We first take a look at results from the pretrain-only setting, in which we directly apply the pretrained models to the test set (it is worth noting that the pretrained training data contains the WMT17 training set). For *single-model*, due to the prohibitively long training time, we are only able to finish two epochs upon the publication of this paper, which has already last 3 months. Though the model has not fully converged, the BLEU score of *single-model* (24.7) is slightly higher than the performance of its model correspondence trained on the WMT17 dataset (24.4).

Mixture models with different dataset split strategies outperform *single-model* by a huge margin. This is due to the reasons that (1) *single-model* has not fully converged yet; (2) model capacities of mixture models are significantly larger (more precisely $K$ times larger) than *single-model*; and most importantly (3) *uniform-data-split* is actually an ensemble of multiple models and the comparison is not completely fair.[4] By comparing *uniform-data-split*, *topic-data-split* and *dynamic-data-split*, we can see that the way in which the full dataset is split has a significant effect on the performance. *topic-data-split* significantly outperforms *uniform-data-split*. This is because subsets in *topic-data-split* contain fewer low-frequency words and sentences with more similar language expression patterns, leading to better performances for models tried on them. *dynamic-data-split* dynamically adjusts training data, which lead to more focused subsets, and thus better performances.

**Pretrain+Finetune** displays similar patterns to *Pretrain-Only*. *single-model* achieves a BLEU score of 28.7, already outperforming the best current system not-ensemble system. The fact that the result of *single-model* was achieved based on the vanilla transformer structure rather than sophisticated structures from the baseline models, and that no additional processing model (e.g., reranking, diverse decoding, back translation, etc) is applied, further demonstrate the significant effectiveness of large-scale pretraining.

The gap between *single-model-1-epoch*, *single-model-2-epoch* and *uniform-data-split* is narrowed in the *Pretrain+Finetune* setting than *Pretrain-Only*. The best setting, *dynamic-data-split* achieves a BLEU score of 32.0.

---

[4] A completely fair comparison would be using an ensemble of 20 *single-model*, each of which is training on the 40B dataset. But this is computationally prohibitive for us.

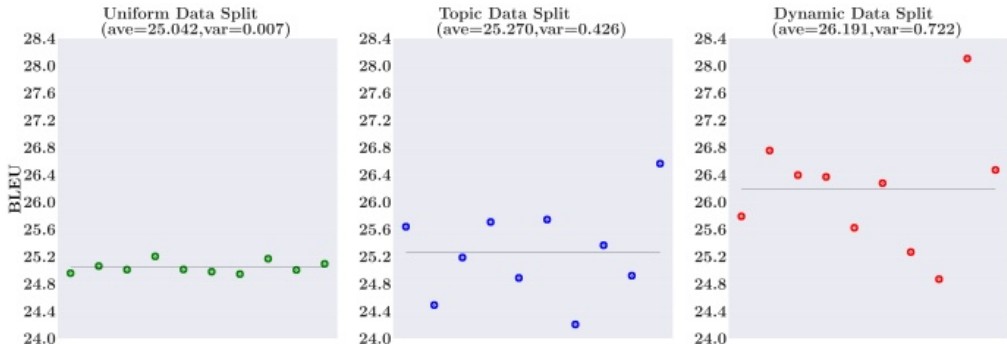

Figure 2: Performans for each of the $K$ components in different data split strategies.

| Model | Dataset | Sufficiently Pretrained | Setting | BLEU |
|---|---|---|---|---|
| Large | Large (40B) | No | Pretrain-Only | 24.7 |
| Large | Small (4B) | Yes | Pretrain-Only | 25.1 |
| Small | Large (40B) | Yes | Pretrain-Only | 23.2 |
| Large | Large (40B) | No | Pretrain+Finetune | 28.7 |
| Large | Small (4B) | Yes | Pretrain+Finetune | 28.4 |
| Small | Large (40B) | Yes | Pretrain+Finetune | 26.8 |

Table 3: Performances for non-ensemble models of different sizes and pre-trained on different sizes of data.

## 6 ABLATION STUDIES AND ANALYSIS

### 6.1 DATA SPLIT STRATEGY

In phrase-based MT, both how the full set is split (Axelrod et al., 2011) and how the mixture models are combined Civera & Juan (2007); Sennrich (2012); Foster et al. (2010) have a significant impact on the final performance. Similar phenomenon is observed in our situation. The first row of Fig 2 corresponds to BLEU scores achieved by each of the $K$ components of the three data-split strategies (pretrain-only). As can be seen, performances for different components of *uniform-data-split* are very similar. This is in accord with our expectation since the dataset is randomly split and all subsets come from the same distribution. Variance scores for *topic-data-split* and *dynamic-data-split* are much larger. This is because subsets of *topic-data-split* and *dynamic-data-split* display a more focused topic/domain distribution. Models trained on more relevant subsets perform much better than models trained on less relevant ones.

### 6.2 MODEL SIZE AND DATA SIZE

It would be interesting to compare the performance of *large-model-large-data*, a large model trained on the full dataset (40B) but has not fully converged, with *large-model-small-data* – a converged large model on a subset (4Billion), and with *small-model-large-data*, – a model of smaller size, with the number of attention heads, embedding dimension and inner-layer dimension set to 8, 512 and 512. The model is run on the full set (40B). Table 3 represents the results. As can be seen, for the pretrain-only and pretrain+finetune setting, *small-model-large-data* performs the worst. When comparing Table 3 with Table 1, *small-model-large-data* even underperforms a larger transformer trained only on the WMT17 dataset. This demonstrates the significant importance of model size and capacity. Interestingly, comparing *large-model-large-data* with *large-model-small-data*, the former performs a bit better on the pretrain-only setting, but performs a bit worse on the pretrain+fine-tune setting. Our explanation is as follows: for the pretrain-only setting, *large-model-large-data* has not fully converged, and performs worse than the fully converged *large-model-small-data* model. But on the pretrain+fine-tune setting, both models are fully converged on the WMT17 dataset. Since *large-model-large-data* saw more data during pretraining process, it has more generalization capability and achieves better performances during fine-tuning.

| Word, BPE and Char | |
|---|---|
| Word | 29.4 |
| BPE | 29.8 |
| Char | 30.1 |

| Different NMT Techniques | |
|---|---|
| Currently best system | 32.02 |
| Monolingual LM Fusion | 30.16 |
| Back Translation | 30.87 |
| Agreement Reranking | 32.22 |
| RL | 32.20 |
| Diverse Decoding | 32.11 |
| Agreement Reranking + RL+ Diverse | 32.29 |

Table 4: Results for word-based, BPE-based and char-based models. (b) results for different existing NMT techniques.

### 6.3 CHARACTER, WORD OR BPE

In large-scale neural network learning, whether Chinese word segmentation (CWS) is still needed is under hot debate (Meng et al., 2019) and the answer is not immediately clear: on one hand, the data sparsity issue with word-based models is less significant when presented with massively large training data. One the other hand, "word" is a human-defined linguistic concept, characterized by labeled CWS datasets (Xia, 2000; Yu et al., 2001). It is widely accepted that large-scale text learning can automatically learn and encode linguistic structures (Hinoshita et al., 2011; Williams et al., 2018), which means CWS might be less useful with large training data. Using the 4B subset on which the best performance is achieved in the *topic-data-split* setting, we use the word-based model (vocab set to 50K), the subword BPE model (vocab set to 50K) and the char-based model (vocab set to 10K). The three models only differ in the encoding stage. Results are shown in Table 4. Combined with fine-tuning, the three models respectively achieve a BLEU score of 29.4, 29.8 and 30.1, validating the finding in (Meng et al., 2019) that CWS is not necessary in NMT when presented with large-scale training data.

### 6.4 EXAMINE OF EXISTING NMT TECHNIQUES

It would be interesting to see which existing widely-used NMT techniques are still effective. Since our bilingual dataset is extremely larger than previous datasets, existing data augmentation strategies such as *monolingual language model fusion* (Sennrich et al., 2015a) or *back translation* (Hassan et al., 2018; Edunov et al., 2018) are expected to be no longer effective. This is because to improve the performance, we need an augmented dataset even larger than our parallel dataset, the training on which is not practical in our case. We verified this hypothesis using 100M monolingual data as shown in Table 4. Other NMT techniques we examined include the following:

Agreement Reranking: Hassan et al. (2018) reranks the N-best list using models that generate sources and targets from different directions, i.e., S2T-L2R (target sentence is generated from left to right), S2T-R2L, T2S-L2R and T2SR2L. Due to the computational cost, we only pretain S2T-R2L, T2S-L2R and T2SR2L on the 4B dataset and then fine-tune them on WMT2017.

Reinforce Learning (Ranzato et al., 2015; Wu et al., 2016): refining the SEQ2SEQ objective with the RL objective to directly optimize the translation BLEU scores. We apply RL in the fine-tuning stage.

Diverse Decoding (Li et al., 2017; Vijayakumar et al., 2018): using a diversity-promoting beam search, in which inter-sibling scores are penalized, to generate more diverse N-best list.

Results are shown in Table 4. As can be since, monolingual ML fusion and back-translation actually harm the performance. This is expected since the incorporated monolingual dataset is significantly smaller than the pretraining bilingual datasets. Agreement reranking introduces +0.17 BLEU boost, validating the effectiveness of the order by which sequences are generated. The improvement from diverse decoding is also tiny, we conjecture that this is because the model is already good enough for beam search to find the global optimal. Another significant improvement comes from the RL strategy, leading to a roughly +0.2 BLEU boost. The combination of RL, agreement-ranking and diverse decoding push the performance up to 32.3.

# 7 CONCLUSION

In this paper, we empirically study training NMT systems with 40B training instances, the size of which is orders of magnitude larger than the largest dataset to date. We design practical resolution to handle the tradeoff between full-dataset level optimal and training speed, and demonstrate that large-scale pretraining can significantly improve NMT performances. We are able to achieve a BLEU score of 32.3 on WMT17 Chinese-English dataset, with a significant performance boost of +3.2 over existing SOTA results

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
