# OpenReview forum: "Large-scale Pretraining for Neural Machine Translation with Tens of Billions of Sentence Pairs"
_ICLR.cc/2020/Conference — Reject_

### Official Review · AnonReviewer1 · 2019-10-20
**Official Blind Review #1**

**Rating:** 3

**Review:**

The paper proposes an approach to train NMT models on extremely large parallel corpora. Because of the dataset size, training several epochs on the full dataset with a single model is too expensive. As a result, the dataset is split into several chunks, on which different models are trained independently. The different models are combined to form an ensemble model. Different strategies are proposed to split the dataset effectively. The resulting model achieves a performance of 32.3, outperforming the previous SOTA by 3.2 BLEU.

In Section 4.3, I'm curious about why only 6 layers are used in the encoder and decoder? Large scale pretraining like in BERT usually benefits from very large datasets, but also very large models. 6 layers seems very small given the size of the training set. Table 2 reports results with "Small" and "Large" models. What does this correspond to? Only Section 4.3 discusses the size of the model, but it does not mention different architectural choices.

In Section 5.1 the paper mentions "the single-model achieves a BLEU score of 29.7, already outperforming the current best system", but in Tables 1 and 2 it seems that the best score with single model is 28.7, not 29.7.

I feel that the results are a bit disappointing given the scale of the experiments. Table 2 suggests that a single model, even trained with 40B sentence pairs, does not outperform a single model trained with 20M sentence pairs as in "He et al., 2019", while being significantly more expensive to train. Also, the comparisons in Table 1 are done between single and ensemble models, which is not a fair comparison. The model with a BLEU score of 32.0 uses an ensemble of 10 models. What would be the performance of an ensemble of 10 models trained with the regular 20M parallel sentence pairs?

Also, did you try the approach of "Hassan et al (2018)" suggested in Section 4.1, where only in-domain sentences are selected? It is true that  "every time we encounter a new domain, we have to retrain the model", but I think this is still a more viable approach than pretraining on the full 40B sentences. Why not trying to train on the top-100M sentence pairs that are the most in-domain?

In the back-translation (BT) experiments, did you select 100M monolingual sentences randomly? If that is the case, this is expected to see a drop in performance, BT is usually a very effective, but not so much when the monolingual data is noisy or out of domain. Although it is critical to work with cleaned data in NLP (especially in the context of generation), dataset cleaning is not really addressed in the paper.

Overall, the experimental setup is impressive, but the improvements in terms of BLEU are relatively small, and the technical contributions seem quite thin to me for a ML conference. Moreover, the dataset used in the paper is not available to the research community, which prevents reproducibility. Also, as mentioned above, I think several important experiments are missing.

**Experience Assessment:**

I have published in this field for several years.

**Review Assessment: Checking Correctness Of Derivations And Theory:**

I carefully checked the derivations and theory.

**Review Assessment: Checking Correctness Of Experiments:**

I carefully checked the experiments.

**Review Assessment: Thoroughness In Paper Reading:**

I read the paper thoroughly.

---

> ### Author Response · Authors · 2019-11-10
> **response to Review #1 (first half)**
>
> Re: I'm curious about why only 6 layers are used in the encoder and decoder? Large scale pretraining like in BERT usually benefits from very large datasets, but also very large models. 6 layers seems very small given the size of the training set.
>
> -- Thank you for the comment. For model structure, we followed the previously SOTA structure of the Microsoft MT system in "Achieving Human Parity on Automatic Chinese to English News Translation". The adopted structured (6 blocks, 16 heads, 1,024 embedding dimension and 2,048 inner-layer dimension) is a commonly used setup in MT. Given fixed memory, the number of layers trades the number of heads, embedding dimension, and batch size. Additionally, the structure for BERT cannot readily be transferred to MT (BERT training uses TPU. And here we use V100).
>
>
> Re: Table 2 reports results with "Small" and "Large" models. What does this correspond to? Only Section 4.3 discusses the size of the model, but it does not mention different architectural choices.
>
> -- Sorry for the confusion. "Small" and "Large" are discussed in the "Model Size and Data Size" paragraph of section 5.  "Small" refers to the model with 6 blocks, 8 heads, 512 embedding dimension and 512 inner-layer dimension and large refers to the model with 6 blocks, 16 heads, 1,024 embedding dimension and 2,048 inner-layer dimension.
>
>
> Re: In Section 5.1 the paper mentions "the single-model achieves a BLEU score of 29.7, already outperforming the current best system", but in Tables 1 and 2 it seems that the best score with single model is 28.7, not 29.7.
>
> -- Nice catch. 29.7 is a typo. All should be 28.7. We will correct it.
>
>
> Re: Table 2 suggests that a single model, even trained with 40B sentence pairs, does not outperform a single model trained with 20M sentence pairs as in "He et al., 2019", while being significantly more expensive to train.
>
> -- Thanks for the comment. We think it is not fair to compare with our result with He et al., 2019 because
> a) He et al., 2019 is an ensemble result
> b) The model that achieves 29.1 involves sophisticated structure design. From their reported results, a vanilla transformer achieves a bleu score of 23.20. Since our result is obtained based on vanilla transformer, it is fair to compare our result with their bleu score  of 23.2, rather than 29.1.
> c) our single model pretrained on 40B ran only 2 epochs (which already took 3 months) and was far from convergence,
>
>
> Re: The comparisons in Table 1 are done between single and ensemble models, which is not a fair comparison. The model with a BLEU score of 32.0 uses an ensemble of 10 models. What would be the performance of an ensemble of 10 models trained with the regular 20M parallel sentence pairs?
>
> -- Sorry for confusion. These results are listed in comparison to our single and ensemble models, respectively.
>
> We choose the following models for comparison to our single models:
> --- Hassan et al., 2018 with a BLEU of 24.2 refers to a single model.
> --- Hassan et al., 2018 with a BLEU of 27.4 in Table 1 refer to a single model
>
> We choose the following models for comparison to our ensemble models:
> --- He et al., 2019 with a BLEU of 29.1 in Table 1 refers to an ensemble model.
> --- Hassan et al., 2018 with a BLEU of 28.4 in Table 1 refers to an ensemble system.
> --- Wang et al., 2017 refers to an ensemble system.
>
> Additionally,
> Our implementation of an ensemble of 10 vanilla transformer models on 20M obtains a bleu score of 26.7, and we will add it to the next version.

---

> > ### Author Response · Authors · 2019-11-10
> > **response to Review #1 (second half)**
> >
> > Re: did you try the approach of "Hassan et al (2018)" suggested in Section 4.1, where only in-domain sentences are selected? It is true that "every time we encounter a new domain, we have to retrain the model", but I think this is still a more viable approach than pre-training on the full 40B sentences. Why not trying to train on the top-100M sentence pairs that are the most in-domain?
> >
> > Thank you for the comment. We actually did try as you suggested and followed Hassan et al (2018) in the first place and found it did not work well. Results from vanilla transformer are shown as follows:
> > top-100M+20M       single           joint training                                            25.4
> > top-100M+20M       single           pretrain(100M)+finetune(20M)             26.2
> > top-100M+20M       ensemble    joint training                                            26.9
> > top-100M+20M       ensemble    pretrain(100M)+finetune(20M)             27.6
> >
> > We will add these results to the updated version.
> > Additionally,
> > 1) Hassan et al (2018) select different training instances for different target domains. This means every time we encounter a new domain, we have to retrain the model.
> > 2) The value of data filtering threshold is crucial but hard to decide.
> >
> > In summary, it is widely accepted that the large-scale training corpus often introduces performance boost. We do believe that using 40B properly will definitely leads to better performance.  The intuition of our work is to leverage the pretraining and fine-tuning pipeline for domain specific translation task. The 40B training corpus is to train a universal machine translator and then fine-tune the model on the training set of the target domain. We believe that
> >
> >
> > Re: In the back-translation (BT) experiments, did you select 100M monolingual sentences randomly? If that is the case, this is expected to see a drop in performance, BT is usually a very effective, but not so much when the monolingual data is noisy or out of domain. Although it is critical to work with cleaned data in NLP (especially in the context of generation), dataset cleaning is not really addressed in the paper.
> >
> > -- It is expected that BT does not help our setting. BT works when we don't have enough parallel data, and we generate significantly larger augmented dataset using BT. But in our setting where we have huge parallel dataset, we need even large BT-generated data to make it help, and this might not be practical.
> > -- As mentioned in Section 3, actually we spent significant amount of time in data cleaning.
> > followed Uszkoreit et al., 2010 `https://storage.googleapis.com/pub-tools-public-publication-data/pdf/36580.pdf` and used a standard dynamic programming approach for data cleaning.
> >
> > Re: The technical contributions seem quite thin to me for a ML conference. The dataset used in the paper is not available to the research community, which prevents reproducibility. As mentioned above, I think several important experiments are missing.
> >
> > -- In this paper, we provide a general mechanism to handle massively large training dataset in NMT. We propose the dynamic data split strategy for pretraining of MT models. This proposed strategy is effective and easy to implement, and can also be applied to any other large-scale pretraining language models as well as other tasks in the field of NLP. We believe such methods can make great contributions to the research community.
> > -- Since the 40B corpus is brought from about 60 commercial translation agencies, we are now actively negotiating with the legal department for the allowance to release the dataset for non-commercial uses.
> >
> >
> > [1] Hassan et al. Achieving Human Parity on Automatic Chinese to English News Translation. 2018
> > [2] He et al. Hard but robust, easy but sensitive: How encoder and decoder perform in neural machine translation. 2019

---

### Official Review · AnonReviewer3 · 2019-10-23
**Official Blind Review #3**

**Rating:** 6

**Review:**


This work conducts a large scale study on pretraining for neural machine translation. Overall, this work makes good contributions to the community, but the experiments need improvements.

Pros:
	1. The data scale is huge, with 40 billion sentence pairs.
	2. The results are promising, with 3.2 BLEU improvement over STOA results.

Cons:
	1. It is pity that the trained model is only evaluated on one test set and experiments are conducted on one language pair. Thus, it is not clear to me whether the improvement is general across datasets and language pairs. I understand that it is costly to conduct such a large scale study on another language pair. At least, it is easy to test the models on other datasets for the same language pair. For example, I'd like to see the results on WMT 2019 Chinese->English translation dataset.
	2. I'm curious how large-scale pretraining compare with large-scale back translation. The following paper shows that large-scale back translation can also significantly improve the final translation accuracy. Note that back translation only needs monolingual data, while the pretraining in this work needs bilingual sentence pairs.
Edunov, Sergey, Myle Ott, Michael Auli, and David Grangier. "Understanding back-translation at scale." arXiv preprint arXiv:1808.09381 (2018).

Besides, will the model be shared to the public?

**Experience Assessment:**

I have published in this field for several years.

**Review Assessment: Checking Correctness Of Derivations And Theory:**

N/A

**Review Assessment: Checking Correctness Of Experiments:**

I carefully checked the experiments.

**Review Assessment: Thoroughness In Paper Reading:**

I read the paper thoroughly.

---

> ### Author Response · Authors · 2019-11-15
> **response to Review #3**
>
> thank you for the useful comments.
>
> 1. regarding WMT19
> Thank you for the sensible comment. We added results on both WMT18 and WMT19, and the model achieves SOTA results on both: a bleu score of 34.4, +2.6 over the best previous system on WMT18 and 42.2, +2.9 over the best previous system on WMT19. Please refer the paper for details.
>
> 2. Large-scale back translation
> Thanks for the comment. We think it is a good idea. We will add more experiments regarding large-scale back translation in the updated version.
>
> 3. data and model publication
>  the corpus is brought from about 60 commercial translation agencies. We are now actively negotiating with the legal department for the allowance to release the dataset for non-commercial uses.

---

### Official Review · AnonReviewer2 · 2019-10-25
**Official Blind Review #2**

**Rating:** 3

**Review:**



This paper investigates the effectiveness of a massively large parallel corpus in NMT training, which consists of more than 40 billion En-Zh parallel sentences.
To the best of my knowledge, the 40 billion parallel corpus for the NMT training is the largest reported in the paper published so far.

For preventing long training time, this paper proposes a practical data split and utilization method, which the authors call “dynamic-data-split.”
The key idea of their method is to dynamically assign training instances to different model components and update different components according to the assigned instances.

This paper reports the BLEU score of WMT17 Chinese-English dataset for 32.3, which significantly outperformed the best score, and improved the performance of existing state-of-the-art results.
They also provide several deeper analyses of the proposed method by changing the model training strategy (pretrain only, pretrain+finetune), data split strategy, data size, and tokenization (word, BPE, character).




The main concern of this paper is the reproducibility of the experiments.

Their main focus is to investigate the effectiveness of 40B massive parallel data.
However, the origin and how the authors correct the data is fully unknown; in the paper, they only say, “The data comes from diverse sources such as web pages (∼2 billion), digitized books (∼1 billion) and private purchase from translation agencies (∼46 billion).”
What is the “private purchase from translation agencies.”
No one knows how they were collected except the authors.
It is impossible to reproduce the results of the experiments conducted in this paper in future validation.



**Experience Assessment:**

I have published in this field for several years.

**Review Assessment: Checking Correctness Of Derivations And Theory:**

I carefully checked the derivations and theory.

**Review Assessment: Checking Correctness Of Experiments:**

I carefully checked the experiments.

**Review Assessment: Thoroughness In Paper Reading:**

I read the paper thoroughly.

---

> ### Author Response · Authors · 2019-11-09
> **response to the Review #2**
>
> thank you for the sensible comments. We do agree that reproducibility is important regarding this paper.
>
> The origin of the dataset: the dataset was purchased from about 60 translation agencies. We do agree that releasing the dataset is of great importance and we are now actively negotiating with the legal department for the allowance to release the dataset for non-commercial uses.
>
> Additionally, this paper for the first time provides a general mechanism to handle massively large dataset in NMT, and this strategy can be easily extended to any large-scale MT datasets.

---

### Public Comment · ~Adam_Bittlingmayer1 · 2019-10-02
**Better evaluation**

These are strong results, and it's a pity to invest so much in machines and then so little in eval.

For the benefit of the research community, I'm happy to offer a free and anonymous ModelFront evaluation.

It would include a break down by error types, actual examples of each error type and the estimated precision and recall against human evaluation.

All you have to do is send a link to system inputs and outputs to eval+Bkl8YR4YDB@modelfront.com.

If you want a comparison between ModelFront eval, human eval and BLEU, then send the reference translations too.

---

### Public Comment · ~Samuel_R._Bowman1 · 2019-10-02
**Minor comments**

Neat! Two minor comments:

- The paper says that "It is widely accepted that large-scale text learning can automatically learn and encode
linguistic structures (Hinoshita et al., 2011; Williams et al., 2018)": I largely agree with your claim, but the citation to our work (Williams et al.) is wrong. In that paper, we argue that a specific class of models *fails* to learn any of the structure that it's meant to learn.

- I'd love to see some example output and qualitative discussion of where you see improvements.

---

> ### Author Response · Authors · 2019-10-03
> **reply**
>
> thanks for the comment. We will correct the incorrect reference.
>
> Also, we will list more example output and do the qualitative analysis in the updated version.

---

### Decision · Program_Chairs · 2019-12-19

**Decision:**

Reject

**Comment:**

The authors address the problem of training an NMT model on a really massive parallel data set of 40 billion Chinese-English sentence pairs, an order of magnitude bigger than other cz-en experiments. To address noise and training time problems they propose pretraining + a couple of different ways of creating a fine-tuning data set. Two of the reviewers assert that the technical contribution is thin, and the results are SOTA but not really as good as you might hope with this amount of data. This combined with the fact that the data set is not released, makes me think that this paper is not a good fit with ICLR and would more appropriate for an application focussed conference. The authors engaged strongly with the reviewers, adding more backtranslation results. The reviewers took their responses into account but did not change their scores.